# Prevalence and factors associated with food insecurity across an entire campus population

**Emilie S. Riddle**[1¤], **Meredith T. Niles**[2]*, **Amy Nickerson**[1]

**1** Department of Nutrition and Food Sciences, The University of Vermont, Burlington, VT, United States of America, **2** Department of Nutrition and Food Sciences & Food Systems Program, University of Vermont, Burlington, VT, United States of America

¤ Current address: Lifestyle Medical Center, Durham, NC, United States of America
* mtniles@uvm.edu

**Data Availability Statement:** All data files are available from the Harvard Dataverse database at https://doi.org/10.7910/DVN/YFYDXU.

**Funding:** The authors received no specific funding for this work.

## Abstract

Comprehensive assessment of food insecurity across all college community members is lacking. This research surveyed a random sample of an entire campus population at a Northeast University in two surveys (spring 2017, n = 1,037 and fall 2017, n = 1,123). Analysis of variance, t-tests, and multivariable logit models were used to understand food insecurity outcomes and comparisons among groups. The overall rate of food insecurity on campus was 19.6% (spring) and 15.0% (fall). Food insecurity rates were highest among undergraduates, graduate and medical students, and staff as compared to faculty. First generation students and off-campus students were also more likely to be food insecure in both surveys, while people of color were more likely to be food insecure in the spring survey. Findings suggest university members beyond undergraduates also face high rates of food insecurity, which has important implications for efforts to reduce food insecurity on college campuses.

## Introduction

Consistent access to a sufficient amount of safe and nutritious food is a fundamental necessity for human health and wellbeing, yet millions of people in the United States experience food insecurity each year. Food insecurity is defined by the USDA as "the limited or uncertain availability of nutritionally adequate and safe foods, or limited or uncertain ability to acquire acceptable foods in socially acceptable ways"; in contrast, hunger is a physiological condition that may result from food insecurity [1]. During 2017, 11.8% of U.S. households experienced food insecurity, with 4.5% experiencing very low food security [2]. Recent evidence highlights that college and university students face higher rates of food insecurity as compared to the national average. Individual studies at four-year colleges and universities have found prevalence rates for low or very low food security that range between 14% to 59% of students, well above the national average [3–13]. However, to date no studies have examined food insecurity rates among all campus populations including graduate and medical students, faculty (defined as professors and lecturers), and staff (other professionals or service workers on campus). To fill this gap in the literature, we surveyed a random sample of an entire campus community at a Northeastern U.S. university to understand prevalence and factors associated with food insecurity among all campus populations.

**Competing interests:** The authors have declared that no competing interests exist.

The factors associated with food insecurity are complex and interrelated, and the prevalence of food insecurity is significantly higher than the national and state averages for some subgroups. Economic factors are most strongly associated with food insecurity, and households with incomes below 185% of the federal poverty level have a food insecurity rate that is three times the national average. Other populations that experience a higher prevalence of food insecurity include households with children (particularly those headed by a single parent), Black and Hispanic households, immigrant households, people living alone, and households that include a person with disabilities[14].

Food insecurity has serious implications for mental and physical health, including deficiencies of essential nutrients or malnutrition, and an increased risk of obesity, chronic diseases, and depression [15]. In a study of over 5000 National Health and Nutrition Examination Survey (NHANES) participants with incomes <200% of the federal poverty level, there was a significant association between food insecurity and clinical evidence of diabetes and hypertension [16]. A significant association between food insecurity and self-report of fair/poor health, along with lower scores on standardized physical and mental health scales, was observed in a cross-sectional study of adults [17]. Children and adolescents living with food insecurity have been found to have a higher risk of poor physical and mental health, including psychosocial and learning outcome impacts, increased BMI and iron deficiency, as well as frequent stomach and headaches and higher rates of hospital admission [15,18]. There is increasing awareness of the imperative need to address food insecurity among post-secondary students, as this population is likely to experience negative academic and health impacts similar to those of children and adults [19].

While there is no annual national survey to assess the prevalence of food insecurity among college students, there have been several large studies and recent systematic reviews of the literature. A recent systematic review of gray and peer-reviewed literature found average rates of food insecurity at postsecondary institutions to be 35% and 42%, respectively [20] and another systematic review of peer-reviewed studies found the rate to be 43.5% [13]. The largest single study to date, conducted by the Wisconsin HOPE Lab in 2018, included over 43,000 respondents who were students at 66 community and four-year colleges in 20 states and Washington D.C, though it was a based on convenience sampling. That survey found that 42% of community college and 36% of four-year college students reported low or very low food security in the previous 30 days [21]. A study of 34 community and four-year colleges, conducted by the College and University Food Bank Alliance (CUFBA) in 2016, included 3,765 respondents in 12 states. That report indicated that 48% of respondents had experienced low or very low food security in the previous 30 days [22].

A growing body of research has identified several factors associated with food insecurity on college campuses. Reflecting national trends, economics were a factor of significance in several studies: students who were employed [3,4,21,22], had loans and higher financial need [6,9,12,22,23], were Pell Grant recipients [11,21], reported housing insecurity or an exogenous economic shock, or were financially independent had higher rates of food insecurity [4,9,12,21]. The CUFBA survey reported that 56% of first generation college students experienced food insecurity, compared to 45% of students whose parents went to college [22]. Having a race/ethnicity other than white, particularly African American or Hispanic, was significantly associated with food insecurity in all studies reviewed with > 350 participants [4,6,7,9,10,12,21–25]. Students living off campus alone or with roommates, or in housing that did not include food provision, experienced higher rates of food insecurity [6,7,10]. However, living on campus and having a meal plan does not prevent food insecurity; the CUFBA and Wisconsin Hope Lab surveys found that 43% and 26% of students with meal plans still experienced food insecurity [21,22]. Associations with gender were not observed in most individual campus studies; one study found that men were significantly more likely to experience food insecurity [10], and two others

noted higher rates among men that were not statistically significant [7,9]. Studies at individual campuses have not assessed rates among trans- or non-gender conforming students; however, the recent Wisconsin HOPE Lab survey found higher rates of food insecurity among women (37%) and students with non-binary gender (46%), as compared to men (28%) [21].

Studies thus far have not consistently assessed the impact of food insecurity on college students' academic performance, mental and physical health, or dietary adequacy. In the CUFBA study, 32% of food insecure students reported academic impacts, including inability to purchase textbooks, and missing or dropping classes [22]. A large study of four public universities in Illinois (n = 1882) and a small study (n = 301) of community college students in Maryland found significant associations between food insecurity and low grade point average [6,24]. In a survey of 58 students utilizing a campus food pantry, 60% of respondents reported adverse academic impacts such as difficulty concentrating in class or during an exam [26]. Further, the recent study of the University of California college campuses found that food insecure students were more likely to have difficulty concentrating and lower academic performance [23]. Students experiencing food insecurity were significantly more likely to report depressive symptoms [4,8,11] or fair/poor physical health [3,4,27]. An earlier Wisconsin HOPE Lab survey of community college students assessed several additional mental health indicators and found higher rates for food insecure as compared to food secure students: 55% reported symptoms of clinical depression (vs. 20%), 52% reported severe anxiety (vs. 18%), 16% identified probable eating disorder symptoms (vs. 5%), and 20% indicated having suicidal thoughts (vs. 6%) [25]. Two studies noted that students had significantly lower intake of fruits and vegetables associated with marginal or low [10] or severe food insecurity [26]. Two studies by Bruening et al., of a sample of college freshman at Arizona State University, included an extensive dietary and physical activity questionnaire and directly measured weight status [8,11]. These studies found that respondents with food insecurity were significantly more likely to report high intakes of fast food and unhealthy eating habits, and less likely to consume breakfast; no associations were observed between dietary quality or weight status [8,11].

The purpose of the current study was to assess the prevalence of food insecurity among all members of a campus community, and identify whether demographic characteristics, including race/ethnicity, gender, first generation status, and off-campus residence are associated with food insecurity. The current survey included undergraduate, graduate and medical students, as well as faculty and staff, to assess food insecurity on campus beyond the undergraduate student experience. While some previous surveys have included graduate students in the sample [3,7,10,23,28], we are aware of only one previously published report that included staff [29], and none that have accounted for faculty community members. We explicitly chose to separate the student population into undergraduate, graduate, and medical students to more completely understand the potential food insecurity challenges of these populations. For example, evidence suggests that on average, graduate students are more likely to be older and have dependents, factors that could affect food security outcomes [30]. Understanding the scope of food insecurity more broadly will add to the growing body of literature related to food insecurity in higher education, inform future campus surveys, and provide context to implement appropriate campus resources and interventions to address food insecurity and the associated negative impacts on community learning, health, and wellbeing.

## Materials and methods

The current study was a repeated cross-sectional analysis of data from two campus food insecurity surveys conducted at a Northeast public university located in an urban/suburban setting, but in a predominantly rural state. We utilize the CHERRIES checklist [31] to report

specific details of our web-based survey. The survey instruments were designed by the Food Insecurity working group at the University, which encompasses faculty, staff, and students, relying upon existing surveys from other institutions and food security validated questions for development. The survey was prepared and distributed by the University's Center for Health and Wellbeing in the spring and fall of 2017. The Institutional Review Board at the University of Vermont approved this study (number 18–0216). Written consent was obtained from all participants. Each web-based survey was sent to a random sample of 4,500 community members, including undergraduate, graduate, and medical students, as well as faculty and staff through email addresses obtained from the University administration. Survey completion was voluntary and all responses were confidential. An individual could only take the survey once, as identified through IP addresses. The first survey was available between February and April 2017; the second survey was available between October and December 2017. As an incentive to increase participation, respondents were entered in a drawing for a $10 gift card. Multiple reminders were sent to non-respondents during the time period of the survey.

Food security status was measured using the validated 10-question U.S. Adult Food Security Survey Module (AFSSM), which measures food security status over the previous 12 months [1]. The questions assess the respondent's experience of the components of food insecurity, including anxiety about food resources, decreased variety and quality of foods in the diet, decreased quantity of food available, and disruptions to eating patterns and intake. The AFSSM questions specify that the condition must occur due to a lack of financial resources, rather than to efforts related to weight control or other factors. Using the USDA guidelines to sum the total affirmative answers for each respondent, a food security level of high, marginal, low, or very low was calculated as shown in Table 1 [32]. The remaining survey questions determined sociodemographic characteristics, including campus group (undergraduate, graduate, medical, faculty, staff), ethnicity, race, gender identity (male, female, trans/non-gender conforming), and number in household. Students were also asked to identify year in school, residence (on- or off-campus) and first-generation college student status. In addition, the second survey identified international students (it was an oversight that this demographic was not included in the first survey). Campus group was self-selected and respondents were only allowed to choose one campus group that they most identified with. Faculty are generally defined as professors (tenured and untenured) as well as lecturers, and staff encompasses a larger group of professionals employed by the university including administration, janitorial services, physical plant services, and non-faculty researchers.

We did not apply any weighting of the data in the analysis. Analysis of variance (ANOVA) were used to assess statistically significant differences between demographic factors with more than two outcomes (i.e. undergraduate, graduate, medical student, faculty and staff as well as class year) and t-tests were used to analyze statistically significant differences for other demographic factors with two outcome variables (e.g. first generation or not). Multivariable logit models were used to understand multiple factors that relate to food insecurity including household size, gender, race/ethnicity (non-white), 1$^{st}$ generation status and on/off campus residence for undergraduate students. For graduate students, faculty and staff, multivariable models

**Table 1. Food security levels based on the U.S. adult food security survey module [1].**

| Affirmative Answers | Food Security Level | Food Security Status | Associated Conditions |
|---|---|---|---|
| 0 | High | Food Secure | No food access issues or limitations |
| 1–2 | Marginal | Food Secure | Anxiety related to food sufficiency |
| 3–5 | Low | Food Insecure | Reduced quality, variety and desirability of diet |
| >5 | Very Low | Food Insecure | Disrupted eating patterns and reduced intake |

included household size, gender, race/ethnicity (non-white), and binary variables for comparing whether a respondent was staff or graduate/medical students. All of these factors were chosen based on previous studies' (as described in the introduction) findings related to the prevalence of food insecurity from these different demographic groups. Results are reported as odds ratios, which can be interpreted as increased or decreased odds of being food insecure. All statistical analyses were performed using Stata 15 (StataCorp, College Station, TX, 2017).

## Results

The completion rate (i.e. number of people who submitted the survey) for the first survey was 24.5%, (+/- 2.94% margin of error, 95% confidence interval); 66 respondents with incomplete surveys were excluded from the final analysis ($n = 1,037$). The response rate for the second survey was 25.8% (+/- 2.82% margin of error, 95% confidence interval); 38 respondents with incomplete surveys were excluded from the final analysis ($n = 1,123$). Individual campus groups total response numbers, campus population at the time, and margins of errors are reported in S1 Table. The prevalence of food insecurity in the total campus sample was 19.6% in the first survey, conducted during the spring semester. This included 11.1% indicating low food security and 8.6% indicating very low food security, with an additional 16.4% of respondents indicating marginal food security. Food insecurity rates were significantly different for campus groups (Table 2), with undergraduates reporting the highest rate of food insecurity at 25.9%. In the second survey, conducted during the fall semester of the following academic year, the prevalence of food insecurity in the total campus sample was 15.0%. This included 7.8% indicating low food security and 7.2% indicating very low food security, with an additional 11.6% of respondents indicating marginal food security. With the exception of the very low food security subgroup, which was just above the threshold for significance, food insecurity rates were significantly different across campus groups in the second survey (Table 2). Graduate and medical students reported the highest rate of food insecurity in the fall survey, 20.8% and 20% respectively, compared to 17.5% for undergraduates.

Table 3 and Fig 1 report food insecurity rates for selected demographic variables. We find statistically significant differences in food insecurity rates among undergraduate, graduate, and medical students and faculty and staff in the spring ($p < 0.001$) and fall survey ($p = 0.001$). In the spring survey, undergraduate food insecurity (25.9%) was significantly more likely compared to faculty food insecurity (1.8%) ($b = -0.241$, $p = 0.000$) and staff food insecurity (13.5%) ($b = -0.123$, $p = 0.001$), whereas graduate student food insecurity (18.6%) was significantly greater compared to faculty food insecurity ($b = -0.168$, $p = 0.017$) (S2a Table). In the fall survey, undergraduate food insecurity (17.5%) ($b = -0.127$, $p = 0.008$) and graduate food insecurity (20.8%) ($b = -0.161$, $p = 0.014$) was significantly greater compared to faculty food insecurity

**Table 2. Food security categorization by campus group for Spring 2017 and Fall 2017 at a Northeast university.**

| Identity | Total Responses (N) | | Food (In)security Categorization (%) | | | | | | | |
| | | | High | | Marginal | | Low | | Very Low | |
| | Spring | Fall | Spring | Fall | Spring | Fall | Spring | Fall | Spring | Fall |
|---|---|---|---|---|---|---|---|---|---|---|
| Undergraduate | 547 | 624 | 54.5 | 68.4 | 19.6 | 14.1 | 13.7 | 9.6 | 12.2 | 7.8 |
| Graduate | 102 | 99 | 60.8 | 67.7 | 20.6 | 11.5 | 8.8 | 10.4 | 9.8 | 10.4 |
| Medical | 49 | 32 | 63.3 | 66.6 | 20.4 | 13.3 | 14.2 | 16.7 | 2 | 3.3 |
| Faculty | 110 | 106 | 94.5 | 92.4 | 3.6 | 2.9 | 0.9 | 3.8 | 0.9 | 1 |
| Staff | 229 | 262 | 76.4 | 79.6 | 10 | 9.4 | 8.3 | 3.5 | 5.2 | 7.5 |

Food insecurity rates overall are the combination of low and very low food security rates.

**Table 3. Rates of food insecurity and varying demographics for Spring 2017 and Fall 2017 at a Northeast four-year university.** ANOVA were utilized with Bonferroni multiple comparison tests (See S2 and S3 Tables) for comparisons with more than one group, while t-tests were used for comparisons with two groups.

| Population | Spring 2017 | | | Fall 2017 | | |
|---|---|---|---|---|---|---|
| | Food Insecurity[a] | n = | p value | Food Insecurity | n = | p value |
| **Undergraduates** | 25.90% | 541 | <0.001 | 17.50% | 561 | 0.001 |
| *1st Year~* | 15.20% | 105 | 0.004 | 11.90% | 134 | 0.019 |
| *2nd Year~* | 23.40% | 158 | | 13.30% | 120 | |
| *3rd Year~* | 31.70% | 145 | | 21.60% | 116 | |
| *4th– 6th Year~* | 30.80% | 133 | | 26.40% | 129 | |
| **Campus Residence** | | | | | | |
| *On Campus* | 19.10% | 261 | 0.018 | 14.10% | 261 | 0.028 |
| *Off Campus* | 27.10% | 428 | | 21.00% | 361 | |
| **First Generation Status[b]** | | | | | | |
| *Non-First Generation* | 21.00% | 572 | <0.001 | 16.50% | 533 | 0.009 |
| *First Generation* | 39.30% | 117 | | 28.10% | 89 | |
| **Other Campus Groups*** | | | | | | |
| *Graduate Students* | 18.60% | 102 | <0.001 | 20.80% | 96 | 0.001 |
| *Medical Students* | 16.30% | 49 | | 20.00% | 30 | |
| *Faculty* | 1.80% | 110 | | 4.70% | 105 | |
| *Staff* | 13.50% | 229 | | 11.00% | 255 | |
| **Race/Ethnicity** | | | | | | |
| *White* | 18.00% | 937 | 0.012 | 14.80% | 864 | 0.769 |
| *People of Color[c]* | 27.80% | 112 | | 15.80% | 107 | |
| **Gender** | | | | | | |
| *Female* | 18.70% | 727 | 0.142 | 14.20% | 663 | 1.000 (compared to male) |
| *Male* | 19.70% | 278 | | 14.20% | 282 | 1.000 (compared to female) |
| *Transgender/Non-binary* | 41.20% | 17 | | 50.00% | 18 | <0.001 (compared to male/female) |

[a] Food insecure defined as low or very low food security

[b] Defined as undergraduate students first in family to attend college

[c] Includes respondents who identified as one or more of Hispanic, American Indian or Alaska native, Asian, Black or African American, Native Hawaiian or Pacific Islander

* Statistically significant differences among undergraduate, graduate, medical, faculty and staff overall. See S2 Table for specific differences between populations.

~ Statistically significant differences among all school years overall. See S3 Table for specific differences between school years.

(4.7%). In both spring (p = 0.004), and fall surveys (p = 0.018) we find significant differences in food insecurity based on class year, with 3rd year student food insecurity (31.7%) (b = 0.165, p = 0.025) and 4th-6th year student food insecurity (30.8%) to a lesser degree (b = 0.148, p = 0.075), significantly greater compared to 1st year food insecurity (15.2%) in the spring survey. In the fall survey, 4th-6th year student food insecurity (26.4%) was significantly greater than 1st year food insecurity (11.9%) (b = 0.144, p = 0.036) (S3 Table). Off campus students (at this institution beyond 1st years) were significantly more like to be food insecure in the spring (27.1%, p = 0.018), and fall (21.0%, p = 0.028) surveys compared to on-campus students. First generation students were significantly more likely to be food insecure in the spring (39.3%, p<0.001) and fall (28.1%, p = 0.009) surveys. People of color were significantly more food insecure (27.8%, p = 0.012) compared to white respondents (18.0%) in the spring survey only. Male and female gender was not associated with food insecurity in either survey; however, respondents who reported transgender or non-binary gender identity had a food insecurity rate of 41.2% in the spring survey (p = 0.142) and 50.0% in the fall survey (p < 0.001).

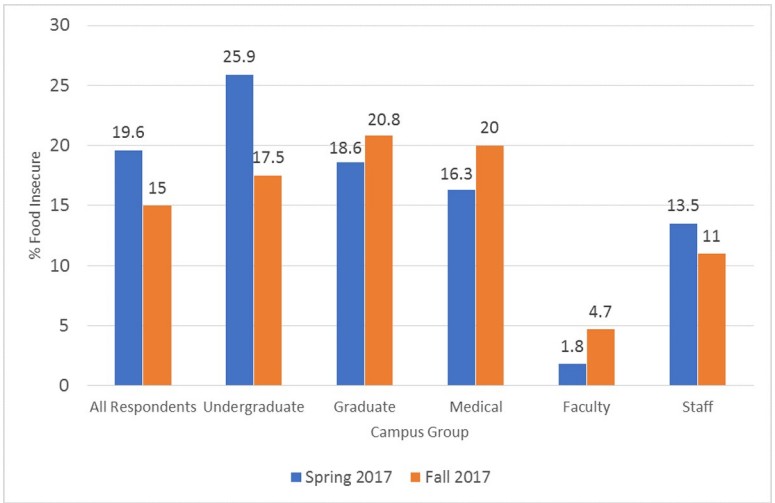

**Fig 1. Food insecurity rates among campus population groups in two surveys.**

In the multivariable analysis of the undergraduate respondents in the spring and fall surveys, first generation college students had more than twice the odds (or = 2.15, $p = 0.01$, or = 2.10, $p = 0.02$) of food insecurity (compared to all other respondents combined) and off campus residents (compared to on-campus) had nearly two times the odds (or = 1.99, $p < 0.01$, or = 1.71, $p = 0.03$) of food insecurity (Tables 4 and 5).

For faculty, staff and graduate/medical students in the spring survey, a larger household size had 1.19 times higher odds ($p = 0.03$) of food insecurity and people of color were twice as likely to experience food insecurity (or = 2.01, p = 0.05). Compared to faculty in the spring survey, staff had more than eight times higher odds of food insecurity (or = 8.08, $p = 0.01$) and graduate/medical students had nearly 13 times higher odds (or = 12.89, $p < 0.01$) of food insecurity. In the fall survey we find less significant increased odds, but higher odds nonetheless, with staff more than two times as likely as faculty to experience food insecurity (or = 2.16, $p = 0.128$) and graduate/medical students having 3.72 times higher odds ($p = 0.02$) of food insecurity compared with faculty (Tables 6 and 7).

## Discussion

This study found an overall campus rate of food insecurity of 19.6% in the spring and 15.0% in the fall for all campus groups. Importantly, and a novel contribution of this work, we find that graduate/medical students and staff, as well as undergraduate students all experienced food insecurity at rates higher than the national average (11.8%) at the time. This finding adds to the growing body of research indicating that food insecurity is more prevalent for undergraduates on college campuses than in the general population, [3,4,6–12,20] but also highlights other

**Table 4. Multivariable analysis of multiple factors related to food insecurity for undergraduate students at a Northeast four-year university, Spring 2017.**

| Variable | Odds Ratio | Std. Error | Z | p = | 95% CI | |
|---|---|---|---|---|---|---|
| Household size | 0.90 | 0.05 | -1.97 | 0.05 | 0.80 | 1.00 |
| Gender | 1.16 | 0.15 | 1.10 | 0.27 | 0.89 | 1.50 |
| Race/Ethnicity | 1.38 | 0.40 | 1.11 | 0.27 | 0.78 | 2.44 |
| 1st Generation | 2.15 | 0.58 | 2.82 | 0.01 | 1.26 | 3.66 |
| Campus Residence | 1.99 | 0.42 | 3.25 | < 0.01 | 1.31 | 3.02 |

**Table 5. Multivariable analysis of multiple factors related to food insecurity for undergraduate students at a Northeast four-year university, Fall 2017.**

| Variable | Odds Ratio | Std. Error | Z | p = | 95% CI | |
|---|---|---|---|---|---|---|
| Household size | 0.94 | 0.08 | -0.80 | 0.42 | 0.80 | 1.10 |
| Gender | 1.40 | 0.20 | 2.34 | 0.02 | 1.06 | 1.85 |
| Race/Ethnicity | 1.36 | 0.53 | 0.80 | 0.42 | 0.64 | 2.92 |
| 1st Generation | 2.10 | 0.68 | 2.29 | 0.02 | 1.12 | 3.97 |
| Campus Residence | 1.71 | 0.42 | 2.18 | 0.03 | 1.06 | 2.78 |

vulnerable groups including graduate and medical students and staff that have been largely overlooked in previous studies.

Most previous published research has focused on undergraduate students, with only two studies reporting rates of food insecurity separately for graduate students [7,10]. One study found food insecurity rates of 16.9% and 5.2% for Master's and PhD students respectively [10]; the other reported a rate of 18% for graduate students. The findings of the current study add additional evidence that students who pursue graduate and medical school continue to have an increased risk, as 18.6% and 20.8% of graduate and 16.3% and 20% of medical students reported experiencing food insecurity in the spring and fall surveys. Economic factors are strongly associated with food insecurity for college students [9] and the general population [14], and graduate and medical students may have additional financial responsibilities or dependent family members. In the spring survey, staff members also indicated a rate of food insecurity (13.5% in the spring, and 11% in the fall) that was higher or equivalent to the national average. Given that there are many types of staff on a university campus, ranging from administrative support to janitorial services, future research should explore in greater detail the vulnerability of specific types of staff to food insecurity.

At the undergraduate level, our results confirm what many others have also found, that university students are frequently more food insecure than the general population, and emphasizes the need to support this vulnerable population to mitigate the negative impacts associated with experiencing food insecurity. While systematic research is needed to quantify the impact on academic performance for college students, food insecurity is detrimental to learning outcomes for children, [18] and it is likely that insufficient access to food also undermines the academic success of college students [6,22,24,26]. Multiple studies have found associations between poorer physical [3,4,27] and mental health [4,8,11,25] and food insecurity; health problems are a clear barrier to academic success, and may prevent students from completing a degree.

Honing in on particular demographics of undergraduate populations may be useful in mitigating food insecurity risks. Our findings suggest that first generation students are of greatest

**Table 6. Multivariable analysis of multiple factors related to food insecurity for graduate/medical students, staff, and faculty[a] at a Northeast four-year university, Spring 2017.**

| Variable | Odds Ratio | Std. Error | Z | p = | 95% CI | |
|---|---|---|---|---|---|---|
| Graduate/Medical[b] | 12.89 | 9.69 | 3.40 | < 0.01 | 2.96 | 56.25 |
| Staff[b] | 8.08 | 6.00 | 2.81 | 0.01 | 1.88 | 34.65 |
| Household size | 1.19 | 0.10 | 2.20 | 0.03 | 1.02 | 1.40 |
| Gender | 0.86 | 0.20 | -0.65 | 0.52 | 0.54 | 1.37 |
| Race/Ethnicity | 2.01 | 0.70 | 1.99 | 0.05 | 1.01 | 3.99 |

[a] Graduate and medical students pooled for multivariable analysis

[b] Compared to faculty

**Table 7. Multivariable analysis of multiple factors related to food insecurity for graduate/medical students, staff, and faculty[a] at a Northeast four-year university, Fall 2017.**

| Variable | Odds Ratio | Std. Error | Z | *p* = | 95% CI | |
|---|---|---|---|---|---|---|
| Graduate/Medical[b] | 3.72 | 2.00 | 2.44 | 0.02 | 1.30 | 10.69 |
| Staff[b] | 2.16 | 1.10 | 1.52 | 0.13 | 0.80 | 5.85 |
| Household size | 0.92 | 0.13 | -0.60 | 0.55 | 0.70 | 1.21 |
| Gender | 1.00 | 0.20 | 0.00 | 1.00 | 0.68 | 1.47 |
| Race/Ethnicity | 0.78 | 0.36 | -0.55 | 0.58 | 0.31 | 1.92 |

[a] Graduate and medical students pooled for multivariable analysis

[b] Compared to faculty

risk (39.3%), which is confirmed by other work such as the CUFBA survey, in which 56% of first generation college students surveyed were food insecure, compared to 45% who had at least one parent who attended college, [22] as well as an earlier Wisconsin HOPE Lab survey that found 43% of first generation respondents were food insecure, compared to 39% of respondents with a parent who attended college. This significantly higher risk may reflect unique vulnerabilities of this population such as access to fewer financial and family support resources.

We also suggest a trend in both surveys among undergraduate students was the "Junior Year Effect," a significant jump in food insecurity in third and fourth, fifth or sixth year students, who are most likely to live off campus, a finding mirrored in previous studies as well [6,7,10]. This increase in food insecurity may be related to the challenges of transitioning off campus, including the lack of meal planning and preparation skills, and the additional financial burdens associated with living independently. It may also be the case that these upper-level students include transfer students from community colleges, which previous research has found to have high rates of food insecurity [24]. Interestingly, although students living on campus are required to have a meal plan at this institution, we also found 19.1% and 14.1% of on campus students were food insecure, reflecting the CUFBA survey data that indicated students with meal plans still experience food insecurity [22]. At most universities, students have the option to purchase smaller meal plans, with the stated expectation that students will purchase or prepare food outside of their meal plan. If students are not supplementing these meal plans, they may be at a higher risk of food insecurity than those on an unlimited meal plan. Further, it is also possible that on-campus students may experience challenges with finding food that is culturally appropriate, meets their allergy or other dietary needs, a concept we intend to explore in previous studies.

Exploring demographic factors across all campus groups elicits further important considerations. First, at least in the spring survey, nearly one-third of people of color in the campus community were food insecure, significantly higher than the average campus rate of 19.6%, and had two times greater odds of experiencing food insecurity. This significant association reflects the national context [14] and trends observed in previous higher education literature [4,6,7,9,10,22,24,25]. Further, respondents who identified as transgender or non-gender conforming had the highest rate of food insecurity observed in this study (41.2% and 50.0% in spring and fall respectively). Due to a very small sample size this result was not statistically significant in the spring semester; however, the rate observed in the fall survey was statistically significant. While this population has not been thoroughly assessed in previous literature on campus food insecurity, the recent Wisconsin HOPE Lab survey found that 46% of students identifying as non-binary gender had experienced food insecurity in the previous 30 days, and

that students identifying as homosexual or bisexual had higher rates of food insecurity than those identifying as heterosexual [21]. Transgender and LGBTQ people may be a population that has less access to support resources, and the unique needs of this population should be considered in designing campus interventions. Household size had different effects across the campus groups. For undergraduate students, particularly for students living off campus, it was associated with reduced odds of food insecurity, perhaps because having roommates may increase available resources for food, and increase the likelihood that a household member has food preparation skills. Conversely, increased household size for graduate and medical students, faculty, and staff resulted in slightly increased odds of food insecurity. Undergraduate students at the institution surveyed are primarily of traditional college age, and the other campus groups included in this survey were more likely to have dependents. It is likely that this finding reflects national data indicating that households with children are more likely to experience food insecurity [14].

Finally, it's worth reflecting on why our results found a lower rate of food insecurity in the fall semester compared with the spring. This finding may be partly due to increased resources at the start of the academic year, such as money earned during the summer break. Additionally, first year students are required to be on an unlimited meal plan for the first semester, increasing the number of students who should have consistent access to food in the fall semester. The difference between the food insecurity rates observed in these surveys suggests the importance of considering the timing of surveys when interpreting research; the current study indicates a possible difference between academic semesters, and a recent study at Arizona State University found significantly higher rates at the end of the semester as compared to the start of the semester [11]. However, another factor that may have contributed to the lower food insecurity rates observed in the second survey was a small change in the survey tool. In the second iteration of the survey a "please explain" box was required if the respondent answered affirmatively to initial screening question number six ("I couldn't afford to eat balanced meals." Was that often, sometimes, or never true for you/your household in the last 12 months?). The extra work involved in answering this question may have driven respondents away from answering affirmatively (see S1 Table). Answering "never true" to this screening question put respondents in the food secure category, possibly contributing to the observed decrease in food insecurity in the fall survey.

Future research is needed to explore factors that contribute to food insecurity among all campus population groups including financial pressures, time of year (e.g., holidays, start of semester, end of semester), availability of preferred or culturally appropriate foods, and food agency, such as meal planning, shopping, and food preparation skills. It is also critical for future research to explore the impact that food insecurity has on academic performance, student retention, and graduation rates. We also suggest that future studies consider surveying their entire campus populations, since these results build upon other studies to demonstrate food insecurity is occurring across all campus populations, especially undergraduate, graduate, medical students and staff. Given the prevalence, it may provide opportunities to streamline efforts to reduce food insecurity and ensure that interventions are available across campus populations.

A limitation of this study was that the questions included in the USDA questionnaire do not indicate the underlying causes related to food insecurity. As a cross-sectional study, the data provides information about a specific point in time, rather than longitudinal data, and causality cannot be assumed. Furthermore, the questions in this survey used to assess food security ask about status during the previous 12 months, so it is possible that periods of food insecurity for students living on campus occur during times when school is not in session, such as holidays and summer break. As such, efforts to reduce food insecurity on college

campuses may not capture these time points. We acknowledge that our survey instrument also failed to capture income levels or sources as well as housing arrangements, which are known predictors of food insecurity.

## Conclusion

Research focused on food insecurity at institutes of higher education has grown significantly, particularly among undergraduate populations. While some studies have also explored graduate students, to our knowledge this is the first study to survey an entire campus population-undergraduate, graduate, and medical students and faculty and staff. These results suggest that one in five members of the community indicated they have experienced food insecurity, adding to the growing evidence that campus communities are more vulnerable to food insecurity than the general U.S. population. Importantly, we find that rates of food insecurity among undergraduate, graduate and medical students as well as staff are also higher than the national average. These results highlight the need for future studies to expand their survey base and consider interventions that include populations outside of undergraduates.

Additional research is needed to quantify the national prevalence of food insecurity on college campuses in a randomized and non-convenience sample method, which could provide comparative data across institutions and provide rigor to the existing national surveys. Furthermore, while many studies are now considering the learning and health outcomes specific to campus communities, these could be expanded, and interventions that are increasingly common on college campuses (e.g. food pantries) could be systematically assessed to understand how effective various strategies may be in supporting campus populations.

However, while necessary, campus and individual level interventions do not address the complex, systems level issues contributing to the prevalence of food insecurity on college campuses. Addressing the root causes of food insecurity in higher education must include a public policy focus on issues such as the increasing cost of education, lack of financial support for college students, rising student debt, housing insecurity, and the insufficiency or inaccessibility of government food and social assistance programs. Helping to ensure that all members of a campus community have consistent access to adequate, nutritious food should be the foundation of quality higher education.

## Supporting information

**S1 Table. Margin of error calculations, total response number and campus population of a given demographic group at the time of the survey.**
(DOCX)

**S2 Table a ANOVA results of food insecurity outcomes by demographic group with Bonferroni multiple comparison tests- Spring 2017 b ANOVA results of food insecurity outcomes by demographic group with Bonferroni multiple comparison tests- Fall 2017.**
(DOCX)

**S3 Table a ANOVA results of food insecurity outcomes by school year with Bonferroni multiple comparison tests- Spring 2017 b ANOVA results of food insecurity outcomes by school year with Bonferroni multiple comparison tests- Fall 2017.**
(DOCX)

## Author Contributions

**Conceptualization:** Meredith T. Niles.

**Data curation:** Meredith T. Niles.

**Formal analysis:** Meredith T. Niles.

**Investigation:** Meredith T. Niles.

**Methodology:** Meredith T. Niles.

**Project administration:** Meredith T. Niles, Amy Nickerson.

**Supervision:** Meredith T. Niles.

**Writing – original draft:** Emilie S. Riddle.

**Writing – review & editing:** Emilie S. Riddle, Meredith T. Niles, Amy Nickerson.

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
