## [Decision Letter · Decision Letter 0]

12 Jun 2020

PONE-D-20-07580

Prevalence and Factors Associated with Food Insecurity Across an Entire Campus Population

PLOS ONE

Dear Dr. Niles,

Thank you for submitting your manuscript to PLOS ONE. After careful consideration, we feel that it has merit but does not fully meet PLOS ONE’s publication criteria as it currently stands. Therefore, we invite you to submit a revised version of the manuscript that addresses the points raised during the review process.

The use of the CHERRIES checklist is important in establishing the limits of the study.

We look forward to receiving your revised manuscript.

Kind regards,

Andrew R. Dalby, PhD

Academic Editor

PLOS ONE

Journal Requirements:

5. Please include your tables as part of your main manuscript and remove the individual files. Please note that supplementary tables (should remain/ be uploaded) as separate "supporting information" files

Additional Editor Comments (if provided):

Reviewers' comments:

Reviewer's Responses to Questions

**Comments to the Author**

1. Is the manuscript technically sound, and do the data support the conclusions?

Reviewer #1: Partly

2. Has the statistical analysis been performed appropriately and rigorously? 

Reviewer #1: Yes

3. Have the authors made all data underlying the findings in their manuscript fully available?

Reviewer #1: Yes

4. Is the manuscript presented in an intelligible fashion and written in standard English?

Reviewer #1: Yes

5. Review Comments to the Author

Reviewer #1: PONE-D-20-07580

Prevalence and Factors Associated with Food Insecurity Across an Entire Campus Population

Thank you for the opportunity to review this paper. The university campus population is emerging as a particularly vulnerable group to food insecurity and as such this paper is adding to the volume of literature on this topic. This paper looks at FI across all staff and students on campus across two time-points Spring and Fall at a US campus. At both times the survey was cross-sectional.

It is a very well written paper and the authors should be congratulated on the clarity of expression. The paper is well-referenced and covers most of the relevant literature in the field. The authors highlight that the novelty of this paper is its focus on faculty and staff – beyond this the paper once again re-iterates prevalence rates and some of the already determined contributing factors. Apart from the staff and faculty data there is very little new reported.

Specific points that would need to be addressed include:

- An international audience will necessarily understand the difference between faculty and staff and this needs to be explained in more depth.

- A rationale about why undergraduate, graduate and medical students need to be distinguished needs to be made.

- To increase rigour use the CHERRIES checklist to report online surveys. Issues such as the rate of completeness, the mechanisms in place to stop a single individual repeatedly undertaking the survey etc

- A total response rate is given against a random sample of 4500 participants who were sent the survey. There is no information on the total population of the university and a more nuanced response rate could be reported indicating the number of faculty/staff that responded, graduate students, medical student, undergraduate students. This would give a better indication of potential bias.

- No data is provided on income levels, income sources, housing arrangements or other aspects that are known to directly impact on FI – instead inference is made for example about living on campus vs off campus etc. In other countries living off campus could be protective as they may be living with family members rather than relying on private rentals. The lack of this data really limits the ability to be able to say anything meaningful about the underlying determinants.

- Demographic data has been collected but there are no new insights provided beyond what has already been reported in the literature. It is important that new studies of this type begin to beyond prevalence rates and begin to seek a more nuanced understanding of determinants and how these can be rectified to prevent FI amongst this group.

6. PLOS authors have the option to publish the peer review history of their article (what does this mean?). If published, this will include your full peer review and any attached files.

Reviewer #1: Yes: Danielle Gallegos

---

## [Author Response · Author response to Decision Letter 0]

20 Jul 2020

Please note the appended below is also included as a file in the file section:

Response to Reviewers

Dear Editors and Reviewers,

 Thank you for the thoughtful feedback on our manuscript, which has improved its overall clarity and rigor. Below we detail our response to specific points in bold.

Reviewer #1: PONE-D-20-07580

Prevalence and Factors Associated with Food Insecurity Across an Entire Campus Population

Thank you for the opportunity to review this paper. The university campus population is emerging as a particularly vulnerable group to food insecurity and as such this paper is adding to the volume of literature on this topic. This paper looks at FI across all staff and students on campus across two time-points Spring and Fall at a US campus. At both times the survey was cross-sectional.

It is a very well written paper and the authors should be congratulated on the clarity of expression. The paper is well-referenced and covers most of the relevant literature in the field. The authors highlight that the novelty of this paper is its focus on faculty and staff – beyond this the paper once again re-iterates prevalence rates and some of the already determined contributing factors. Apart from the staff and faculty data there is very little new reported.

Specific points that would need to be addressed include:

1. An international audience will necessarily understand the difference between faculty and staff and this needs to be explained in more depth.

Response: Excellent point, thank you. We have added language to further distinguish this in the introduction briefly and in the methods more thoroughly.

- A rationale about why undergraduate, graduate and medical students need to be distinguished needs to be made.

Response: We have added additional clarification and rational for separating out these three types of students in the introduction.

- To increase rigour use the CHERRIES checklist to report online surveys. Issues such as the rate of completeness, the mechanisms in place to stop a single individual repeatedly undertaking the survey etc

Response: Thank you, we have included some additional details in the methods section related to the CHERRIES checklist.

- A total response rate is given against a random sample of 4500 participants who were sent the survey. There is no information on the total population of the university and a more nuanced response rate could be reported indicating the number of faculty/staff that responded, graduate students, medical student, undergraduate students. This would give a better indication of potential bias.

Response: The information requested was provided in Supplementary Table 1, which is copied below for reference. Perhaps this was inaccessible to the reviewer, hopefully this is the information that they were seeking.

Supplementary Table 1. Margin of error calculations, total response number and campus population of a given demographic group at the time of the survey.

 Spring 2017 Fall 2017

Demographic Group Margin of Error Response number Campus Population Margin of Error Response number Campus Population

Undergraduate 4.07% 547 9786 3.81% 624 10513

Grad 9.35% 102 1406 9.53% 99 1517

Med 13.24% 49 457 16.73% 32 459

Faculty 9.02% 110 1600 9.20% 106 1600

Staff 6.16% 229 2373 5.71% 262 2373

Overall 2.94% 1037 15622 2.82% 1123 16462

- No data is provided on income levels, income sources, housing arrangements or other aspects that are known to directly impact on FI – instead inference is made for example about living on campus vs off campus etc. In other countries living off campus could be protective as they may be living with family members rather than relying on private rentals. The lack of this data really limits the ability to be able to say anything meaningful about the underlying determinants.

Response: We did not inquire about income level or sources among the respondents, nor housing arrangements, beyond understanding whether or not undergraduate students were living on or off campus. We appreciate this comment, as we recognize that this does limit our understanding and interpretation of results and would be a useful addition to future surveys and research. We have added this as a limitation of the study in the discussion section. 

- Demographic data has been collected but there are no new insights provided beyond what has already been reported in the literature. It is important that new studies of this type begin to beyond prevalence rates and begin to seek a more nuanced understanding of determinants and how these can be rectified to prevent FI amongst this group.

Response: We agree with this assessment, and have been working in our Food Insecurity Working Group on campus to implement many strategies for food insecurity alleviation, and examining the impact of these interventions. Additional analysis also undergoing on the second survey further explores the role of food agency in food insecurity.

---

## [Editor Report · Decision Letter 1]

31 Jul 2020

Prevalence and Factors Associated with Food Insecurity Across an Entire Campus Population

PONE-D-20-07580R1

Dear Dr. Niles,

We’re pleased to inform you that your manuscript has been judged scientifically suitable for publication and will be formally accepted for publication once it meets all outstanding technical requirements.

Kind regards,

Andrew R. Dalby, PhD

Academic Editor

PLOS ONE
---

## [Editor Report · Acceptance letter]

18 Aug 2020

PONE-D-20-07580R1 

Prevalence and Factors Associated with Food Insecurity Across an Entire Campus Population 

Dear Dr. Niles:

I'm pleased to inform you that your manuscript has been deemed suitable for publication in PLOS ONE. Congratulations! Your manuscript is now with our production department. 

Kind regards, 

on behalf of

Dr. Andrew R. Dalby 

Academic Editor

PLOS ONE